# Algorithm Aversion as an Obstacle in the Establishment of Robo Advisors

**Ibrahim Filiz [1,*] , Jan René Judek [1] , Marco Lorenz [2] and Markus Spiwoks [1]**

1   Faculty of Business, Ostfalia University of Applied Sciences, Siegfried-Ehlers-Str. 1,
    D-38440 Wolfsburg, Germany
2   Faculty of Economic Sciences, Georg August University Göttingen, Platz der Göttinger Sieben 3,
    D-37073 Göttingen, Germany
*   Correspondence: ibrahim.filiz@ostfalia.de

**Abstract:** Within the framework of a laboratory experiment, we examine to what extent algorithm aversion acts as an obstacle in the establishment of robo advisors. The subjects had to complete diversification tasks. They could either do this themselves or they could delegate them to a robo advisor. The robo advisor evaluated all the relevant data and always made the decision which led to the highest expected value for the subjects' payment. Although the high level of efficiency in the robo advisor was clear to see, the subjects only entrusted their decisions to the robo advisor in around 40% of cases. In this way, they reduced their success and their payment. Many subjects orientated themselves towards the 1/n-heuristic, which also contributed to their suboptimal decisions. As long as the subjects had to make decisions for others, they noticeably made a greater effort and were also more successful than when they made decisions for themselves. However, this did not have an effect on their acceptance of robo advisors. Even when they made decisions on behalf of others, the robo advisor was only consulted in around 40% of cases. This tendency towards algorithm aversion among subjects is an obstacle to the broader establishment of robo advisors.

**Keywords:** algorithm aversion; robo advisors; decisions for others; portfolio choice; diversification; behavioural finance; experiments

**JEL Classification:** D81; D84; D91; G11; G21; G41; O31; O33

## 1. Introduction

The traditional portfolio management business is demanding in terms of human resources and therefore comparatively expensive. Wealthy private customers have, however, become more price-sensitive since the establishment of low-cost investment opportunities such as exchange-traded funds (ETFs) in recent decades. Many banks are thus trying to find low-cost alternatives, particularly for the support of customers with smaller and medium-sized assets. The increased use of automated processes in portfolio management offers considerable scope for cost reduction. Many banks thus offer robo advisors (see, for example, Rühr et al. 2019a; Jung et al. 2018; Singh and Kaur 2017). Robo advisors are algorithms which are specialised in making investment decisions for customers and processing them. Using new technologies such as artificial neural networks, robo advisors are becoming increasingly more powerful and can potentially maximise clients' returns (Méndez-Suárez et al. 2019).

However, many customers have reservations about interacting with automated processes (robo advisors), although the latter are often remarkably effective (see, for example, Rossi and Utkus 2020; Bhatia et al. 2020; D'Acunto et al. 2019; Beketov et al. 2018; Uhl and Rohner 2018). So-called algorithm aversion is thus a significant problem for the banking sector.

Algorithm aversion particularly occurs when algorithms have to deal with stochastic processes. This is undoubtedly the case with robo advisors. Even when the algorithm makes very good investment decisions, it will—given the stochastic nature of financial market trends—never be able to always make perfect investment decisions. Dietvorst et al. (2015) showed that the tolerance of occasional errors by algorithms is much lower than the tolerance shown regarding occasional poor decisions which one has taken oneself or are made by an expert. We speak of algorithm aversion when subjects decline the use of an algorithm even though it is clearly recognisable that their own decisions or those of experts are by no means more successful (for the usual definitions, see, for example, Filiz et al. 2021a). There is a considerable amount of research results available on measures which can mitigate algorithm aversion (see, for example, Hinsen et al. 2022; Filiz et al. 2021b; Gubaydullina et al. 2021; Kim et al. 2021; Jung and Seiter 2021; Castelo et al. 2019; Dietvorst et al. 2018; Taylor 2017).

The efficiency of robo advisors is due—among other things—to the fact that they can make meaningful diversification decisions effortlessly. By contrast, investors often find it difficult to determine the expected earnings and the risk (variance) of alternative investments and to take into account the correlations of different investment opportunities in an appropriate way (see, for example, Ungeheuer and Weber 2021; Cornil et al. 2019; Enke and Zimmermann 2019; Gubaydullina and Spiwoks 2015; Eyster and Weizsäcker 2011; Kallir and Sonsino 2009; Hedesstrom et al. 2006). This is why in practice many portfolios prove to be under-diversified or diversified in unsuitable ways (see, for example, Gomes et al. 2021; Chu et al. 2017; Dimmock et al. 2016; Anderson 2013; Hibbert et al. 2012; Goetzmann and Kumar 2008; Meulbroek 2005; Polkovnichenko 2005; Huberman and Sengmueller 2004; Agnew et al. 2003; Guiso et al. 2002; Benartzi 2001; Benartzi and Thaler 2001; Barber and Odean 2000; Bode et al. 1994; Blume and Friend 1975; Lease et al. 1974).

We build on studies that have examined what influences the willingness to use a robo advisor. Alemanni et al. (2020) showed that the willingness to follow a robo advisor is lower when the robo advisor suggests a portfolio change. If the current portfolio is to be retained, the willingness to use is similar to advice from human advisors (Alemanni et al. 2020). In a questionnaire-based study, von Walter et al. (2021) found that consumers who believe artificial intelligence is better than human intelligence are more likely to accept advice from a robo advisor. Hodge et al. (2021) showed that subjects follow advice from a robo advisor without a name more closely than advice from a robo advisor that has been given a name. Robo advisors with names tend to be more popular for simple tasks than for complex ones (Hodge et al. 2021). The age of the decision-maker may also be a factor: Robillard (2018) argued that millennials may rely more heavily on robo advisors because this generation has lower trust in fellow humans than other generations.

Users' risk attitudes and attitudes toward automated processes also influence robo advisors and their investment decisions. Robo advisors can identify different risk profiles of their users, although there are large differences in risk preferences within the same investor type group (Boreiko and Massarotti 2020). User preferences, however, have different effects on the perception of and intention to use robo advisors. For financial investments, a higher perceived level of automation leads to higher performance expectations and higher user control leads to lower perceived risk (Rühr et al. 2019b). Since robo advisors should take user preferences into account to increase their usage intent, a performance-control dilemma arises that needs to be mitigated (Rühr 2020).

Another important aspect seems to be the transfer of responsibility to the robo advisor. Niszczota and Kaszás (2020) discovered that moral investment decisions are rather delegated to humans than to robo advisors. On the other hand, Back et al. (2021) showed that subjects feel better in cases of loss if they have delegated some of the responsibility to the robo advisor. For tasks outside the world of finance, it has already been shown that punishment by third parties can be significantly lower if errors are committed by an algorithm rather than by a human (Feier et al. 2022). The idea that, under certain circumstances,

subjects may be happy to hand over responsibility for possible future mistakes to a robo advisor is remarkable, and we explore it in more detail in the present study.

We carried out an economic experiment in which the subjects had to make four investment decisions. They could choose between different investment alternatives in each of the four cases. They were informed of the possible returns, the probability that these returns would materialise, and the correlations of the different investment opportunities. The subjects could either make their own diversification decisions or entrust the task to a robo advisor. The subjects knew that the robo advisor took all of the relevant data into account (the expected value of the returns, the probability that the returns would materialise, and the correlation coefficients of the return development of the different investments), evaluated them optimally, and took them into account in its investment decisions. However, the subjects were also aware of the fact that the robo advisor could not know which random event will occur next. The subjects received the risk-adjusted return of their investment decisions as payment. This had the advantage that the subjects' risk preferences had no meaning for the assessment of the investment alternatives. We examine whether algorithm aversion occurs in this context and whether this can lead to a reduction in risk-adjusted returns. In this context, adding to previous research, we also consider whether algorithm aversion is less pronounced when a person has to make decisions for others.

Some empirical research findings indicate that when making decisions for others a change in the willingness of subjects to take risks can come into play (see, for example, Andersson et al. 2022; Eriksen et al. 2020; Vieider et al. 2016; Pahlke et al. 2015; Füllbrunn and Luhan 2015; Bolton et al. 2015; Pahlke et al. 2012; Chakravarty et al. 2011; Charness and Jackson 2009; Reynolds et al. 2009). This is particularly true when the person for whom a decision is being made is actually present (Polman 2012). Later on, the persons for whom a decision is made may demand that the decision-maker justify their choices. If this is known in advance, it can lead to particular care on the part of the decision-maker. If the decision is delegated to an algorithm, however, the decision-makers do not have to justify their choices. This could possibly contribute towards a reduction in algorithm aversion.

This study examines the circumstances under which robo advisors can become an important complementary tool in wealth management. In addition to the performance of robo advisors in the meaningful diversification of investment alternatives, the reluctance of subjects to use automated processes (in this case: robo advisors) is the focus of attention. Measures to dampen algorithm aversion are of considerable interest. In this context, we raise the question of whether people are more likely to use a robo advisor if the consequences of the robo advisor's decision affect third parties. Exploring this question can help reduce hurdles to establish robo advisors. It also contributes to our understanding of the relatively new research field of algorithm aversion. The rest of this research paper is organised as follows: In Section 2, the experimental design is explained. Section 3 deals with the elaboration of the hypotheses. In Sections 4 and 5, the results of the economic experiments are presented and discussed in the context of the existing literature. To wrap up this study, Section 6 provides a summary of the key findings and a conclusion.

## 2. Experimental Design

In order to answer the research question, an economic experiment with two treatments was carried out between 20 and 28 April 2022 in the Ostfalia Laboratory of Experimental Economic Research (OLEW) at Ostfalia University of Applied Sciences in Wolfsburg. A total of 160 students of the Ostfalia University of Applied Sciences took part in the experiment. Of these, 112 subjects (70%) were male and 48 subjects (30%) were female. Of the 160 participants, 98 subjects (61.25%) studied at the Faculty of Economics and Business, 38 subjects (23.75%) at the Faculty of Vehicle Technology, and 24 subjects (15%) at other faculties. Their average age was 23.6 years.

In each treatment, subjects have to make four investment decisions (tasks 1–4) whose success directly affects them (or others). However, the subjects do not profit from gains

in the share prices—they only profit (once) from the dividend payments of the shares in 2022. The subjects can either make their own diversification decisions or entrust the task to a robo advisor. In the treatment entitled 'Self' the subjects make a diversification decision for their own portfolio and receive the payment themselves. In the treatment entitled 'Representative' the subjects make a diversification decision for another participant's portfolio and the other participant in the session receives the payment which has been obtained. In the treatment 'Representative', after the payment has been made, the subjects are informed about who is responsible for which payment.

Let us assume, for example, that subject C receives the payment achieved by subject B and vice versa (Figure 1). After the experiment, subject B could demand in a personal conversation that subject C justifies his or her decisions. Moreover, subject C could also demand that subject B justifies their decisions. All of the subjects who participate in the treatment 'Representative' are informed about this at the beginning of the experiment.

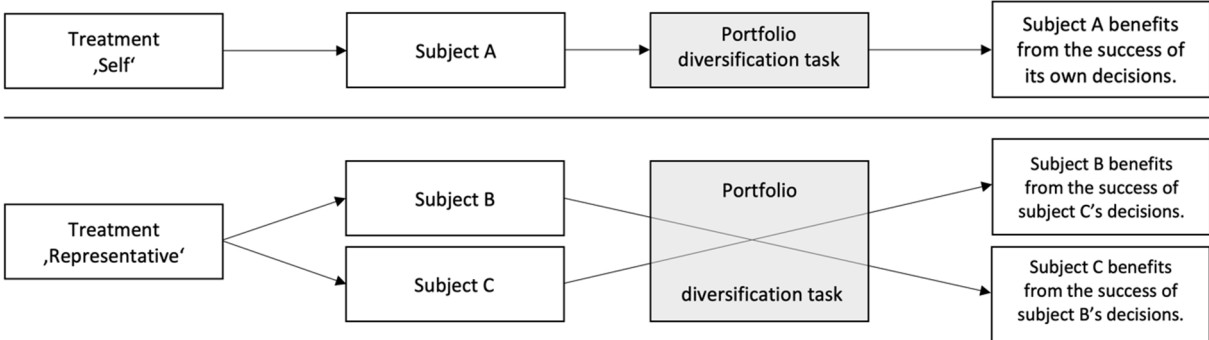

**Figure 1.** The treatment 'Self' and the treatment 'Representative'.

In the first task, there are two shares to choose from: share Y and share Z. The dividend payments of both companies are independent random processes with two possible configurations: 8 experimental currency units (ECU) and ECU 0. The probability of each of these occurring is 50%. The expected values of the dividend payments are thus ECU 4 each. The dividend payments of the two shares are wholly uncorrelated (correlation coefficient = 0). Table 1 shows the level of the dividend payments of the two shares in the past ten years. In this task, as well as in all other tasks in the experiment, subjects are given the dividend payments for the years 2012 to 2021. The dividend payments for 2022, which are relevant for their payoff, are still unknown, which is illustrated by the question mark.

**Table 1.** History of the random events of the dividend payments in task 1.

|         | 2012  | 2013  | 2014  | 2015  | 2016  | 2017  | 2018  | 2019  | 2020  | 2021  | 2022 |
|---------|-------|-------|-------|-------|-------|-------|-------|-------|-------|-------|------|
| Share Y | ECU 8 | ECU 0 | ECU 8 | ECU 8 | ECU 8 | ECU 0 | ECU 8 | ECU 0 | ECU 0 | ECU 0 | ?    |
| Share Z | ECU 0 | ECU 0 | ECU 8 | ECU 0 | ECU 8 | ECU 8 | ECU 0 | ECU 0 | ECU 8 | ECU 8 | ?    |

The subjects are allowed to compile a portfolio consisting of two shares. They can thus choose two Y shares, two Z shares, or one Y share and one Z share. As payment, they receive the risk-adjusted dividends for 2022. A risk-adjusted dividend is equivalent to the dividend payment divided by the variance of the dividend payments of the chosen portfolio. The task thus consists of achieving the highest possible dividends with the lowest possible risk (low variance). The total of all risk-adjusted dividends (in ECU) which are obtained via portfolio decisions is multiplied by five at the end and then paid in euros.

As the subjects do not know the next random events for the dividend payments of share Y and share Z, it makes sense for them to orientate themselves towards the expected values and the variances of the three possible portfolios (see Table 2).

**Table 2.** Expected values and variances in task 1.

| Possible Portfolios | Expected Value of the Dividend | Variance | Expected Value of the Payment |
|---|---|---|---|
| 2 Y shares | ECU 8 | 64 | ECU 0.125 or EUR 0.625 |
| 2 Z shares | ECU 8 | 64 | ECU 0.125 or EUR 0.625 |
| 1 Y share + 1 Z share | ECU 8 | 32 | ECU 0.25 or EUR 1.25 |

Rational economic subjects orientate themselves towards the expected values of the payment, i.e., they select the mixed securities portfolio (1 Y share + 1 Z share). This is exactly how the robo advisor works.

All of the subjects have been familiarised with stochastic processes and the calculation of probabilities at school and also at the beginning of their degree programmes. They are aware of the fact that one cannot draw any conclusions about future random occurrences from an independent random event. Nevertheless, the temptation is great to make a forecast on which events will occur in the cases of the two shares in 2022 which is derived from the sequence of favourable and unfavourable dividend payments. People tend to see patterns even where there are definitely none (see, for example, Zielonka 2004; Wärneryd 2001; Gilovich et al. 1985; Roberts 1959). Subjects who have succumbed to the hot hand fallacy (Burns 2001; Gilovich et al. 1985) will tend to choose the portfolio of 2 Z shares. Subjects who believe in the gambler's fallacy (Rogers 1998; Tversky and Kahneman 1971) will prefer the 2 Y shares portfolio. Subjects who think they can predict the next random events will not make use of the robo advisor. Subjects who want to maximise the expected value of their payment can, however, sleep easily if they delegate the decision to the robo advisor because the robo advisor is specialised in making meaningful portfolio decisions and takes all of the relevant information into account in an optimal way in order to achieve risk-adjusted dividend payments which are as high as possible. The subjects are informed of this.

The second task is somewhat more complex. Once again, there are two shares to choose from (share X and share Q). Both of the shares can pay a dividend of either ECU 4 or ECU 0. The probability of each of these occurring is 50%. The expected values of the dividend payments are thus ECU 2 each. Once again, they are independent random events. The dividend payments of share X and share Q are completely uncorrelated (correlation coefficient = 0).

Table 3 shows the level of the dividend payments of the two shares in the last 10 years.

**Table 3.** History of the random events of the dividend payments in task 2.

|  | 2012 | 2013 | 2014 | 2015 | 2016 | 2017 | 2018 | 2019 | 2020 | 2021 | 2022 |
|---|---|---|---|---|---|---|---|---|---|---|---|
| Share X | ECU 0 | ECU 0 | ECU 4 | ECU 0 | ECU 0 | ECU 0 | ECU 4 | ECU 4 | ECU 4 | ECU 4 | ? |
| Share Q | ECU 0 | ECU 4 | ECU 4 | ECU 4 | ECU 0 | ECU 4 | ECU 0 | ECU 0 | ECU 4 | ECU 0 | ? |

The subjects can compile a portfolio consisting of four shares. They can thus choose four X shares, four Q shares, three X shares and one Q share, three Q shares and one X share, or two X shares and two Q shares. Neither the subjects nor the robo advisor know what the random events (dividend payments for share X and share Q) will be in 2022. A rational subject would orientate themselves towards the expected value of the payment and select the portfolio 2 X shares + 2 Q shares (see Table 4). This is exactly what the robo advisor does.

**Table 4.** Expected values and variances in task 2.

| Possible Portfolios | Expected Value of the Dividend | Variance | Expected Value of the Payment |
|---|---|---|---|
| 4 X shares | ECU 8 | 64 | ECU 0.125 or EUR 0.625 |
| 4 Q shares | ECU 8 | 64 | ECU 0.125 or EUR 0.625 |
| 3 X shares + 1 Q share | ECU 8 | 40 | ECU 0.20 or EUR 1 |
| 3 Q shares + 1 X share | ECU 8 | 40 | ECU 0.20 or EUR 1 |
| 2 X shares + 2 Q shares | ECU 8 | 32 | ECU 0.25 or EUR 1.25 |

The third task and the fourth task can no longer be accomplished with a crude diversification strategy such as the 1/n heuristic (see, for example, Fernandes 2013; Baltussen and Post 2011) because these are companies which belong to the same industry sector and whose dividend payments depend on the success of the sector. The dividend payments of the two shares are thus completely positively correlated (correlation coefficient = 1). Table 5 shows the amount of the dividend payments in the past ten years.

**Table 5.** History of the random events of the dividend payments in task 4.

|  | 2012 | 2013 | 2014 | 2015 | 2016 | 2017 | 2018 | 2019 | 2020 | 2021 | 2022 |
|---|---|---|---|---|---|---|---|---|---|---|---|
| Share M | ECU 4 | ECU 0 | ECU 4 | ECU 0 | ECU 0 | ECU 0 | ECU 4 | ECU 4 | ECU 0 | ECU 4 | ? |
| Share P | ECU 3 | ECU 1 | ECU 3 | ECU 1 | ECU 1 | ECU 1 | ECU 3 | ECU 3 | ECU 1 | ECU 3 | ? |

A phase in which companies in this sector are either successful or are struggling occurs purely coincidentally with a probability of 50%. Previous events thus provide no indication of which random events might occur in the future. The expected value of the dividend payments is thus ECU 2 for both shares. The subjects can compile a portfolio consisting of four shares.

Given that the dividend payments for both shares are 100% positively correlated, a mixture of the two shares does not create any diversification effect. The optimal strategy is to select four P shares because that is the minimum variance portfolio (see Table 6). This is precisely the strategy pursued by the robo advisor.

**Table 6.** Expected values and variances in task 4.

| Possible Portfolios | Expected Value of the Dividend | Variance | Expected Value of the Payment |
|---|---|---|---|
| 4 M shares | ECU 8 | 64 | ECU 0.125 or EUR 0.625 |
| 4 P shares | ECU 8 | 16 | ECU 0.50 or EUR 2.50 |
| 3 M shares + 1 P share | ECU 8 | 49 | ECU 0.165 or EUR 0.825 |
| 3 P shares + 1 M share | ECU 8 | 25 | ECU 0.32 or EUR 1.60 |
| 2 M shares + 2 P shares | ECU 8 | 36 | ECU 0.225 or EUR 1.125 |

The experiment proceeds as follows: First, the subjects read the instructions and answer the control questions (see Appendices A and B). Afterwards, they make the four portfolio decisions of tasks 1 to 4 either with the help of the robo advisor or independently (see Appendix C). For each of the four tasks, the subjects can decide again whether they want to delegate the task to the robo advisor or whether they want to choose a portfolio composition themselves. Only after the four tasks have been completed is it revealed which random events have occurred in this session and to which compensation the subjects have progressed. The payment is then made in cash.

### 3. Hypotheses

The most meaningful strategy is to delegate all four tasks to the robo advisor. The robo advisor always makes the most meaningful decisions. It always selects the portfolio

composition which maximises the expected value of the payment in euros. It would actually be possible to work out this optimal decision oneself. However, the amount of effort required to do so is considerable. The subjects can make mistakes when calculating the expected payment amount. The robo advisor, on the other hand, always evaluates all of the relevant data in an optimal way and always makes the decision which maximises the expected value of the payment. Nevertheless, it has to be expected that some subjects will have reservations about using a robo advisor. The wide variety of previous findings on the occurrence of algorithm aversion make this highly likely (Mahmud et al. 2022; Kawaguchi 2021; Burton et al. 2020; Castelo et al. 2019; Prahl and Van Swol 2017).

**Hypothesis 1.** *Not all of the subjects will trust the robo advisor (algorithm), although it is not possible for them to make a better decision. This means that algorithm aversion will occur.*

**Null Hypothesis 1.** *All of the subjects will trust the robo advisor (algorithm). This means that algorithm aversion will not occur.*

If the subjects are wary of using the robo advisor (algorithm aversion), this may well lead—on average—to a reduction in the payment they obtain. Algorithm aversion will presumably cause a loss in potential earnings.

**Hypothesis 2.** *The more frequently the subjects delegate their decision to the robo advisor, the higher their payments will be.*

**Null Hypothesis 2.** *The frequency with which the subjects delegate their decisions to the robo advisor does not have a positive influence on their payment.*

Among the subjects, there will presumably be some who pursue a crude diversification strategy (1/n-heuristic; see, for example, Fernandes 2013; Morrin et al. 2012; Baltussen and Post 2011; Huberman and Jiang 2006; Benartzi and Thaler 2001). This strategy can lead to success in tasks 1 and 2. In tasks 3 and 4, on the other hand, it cannot lead to success. For an optimal solution of tasks 3 and 4, it is necessary to also take into account the correlation coefficients alongside the expected values of the dividends.

**Hypothesis 3.** *Subjects who do not deploy the algorithm partly neglect the correlations, and in the cases of tasks 3 and 4 they find the optimal solution significantly less often than in tasks 1 and 2.*

**Null Hypothesis 3.** *Subjects who do not deploy the algorithm do not neglect the correlations, and in the cases of tasks 3 and 4 they do not find the optimal solution significantly less often than in tasks 1 and 2.*

On the basis of the existing research on decision making for others (see, for example, Pahlke et al. 2015; Polman 2012; Pahlke et al. 2012; Charness and Jackson 2009; Reynolds et al. 2009) we presume that the subjects who make decisions for others (the treatment 'Representative') consider their decisions more carefully and try harder to make meaningful decisions. After all, the persons for whom the decisions are being made are actually present. At the end of the experiment, who decided for whom and what the results were is announced. All of the subjects in the treatment 'Representative' are aware of this. In other words, they have to expect that they will need to justify their decisions. The subjects in the treatment 'Self', on the other hand, are only responsible for themselves. They need not fear that someone will demand that they justify their decisions. We therefore presume that algorithm aversion will occur less frequently in the treatment 'Representative' than in the treatment 'Self'. In addition, we presume that those persons in the treatment 'Representative' who do not want to trust the robo advisor—for whatever reason—will make a greater effort to select meaningfully diversified portfolios.

**Hypothesis 4.** *The solution of the tasks is delegated to the robo advisor significantly more often in the treatment 'Representative' than in the treatment 'Self'.*

**Null Hypothesis 4.** *The solution of the tasks is not delegated to the robo advisor significantly more often in the treatment 'Representative' than in the treatment 'Self'.*

**Hypothesis 5.** *Those persons who do not want to trust the robo advisor will choose the optimal portfolio structure significantly more often in the treatment 'Representative' than in the treatment 'Self'.*

**Null Hypothesis 5.** *Those persons who do not want to trust the robo advisor will not choose the optimal portfolio structure significantly more often in the treatment 'Representative' than in the treatment 'Self'.*

The general research question of this study is: Can robo advisors become useful complementary tools in the modern wealth management business? In order to explore our research question, we assume that a robo advisor cannot forecast future capital market developments without errors. However, a robo advisor can effortlessly make meaningful diversification decisions. This leads to the question if economic agents need a robo advisor in order to achieve good diversification decisions with certainty. In four very clear decision situations where shares are assembled into a portfolio, optimal decisions can easily be made. However, the facts (expected value, the dispersion of events around expected value, and the correlation of the events of different shares) are neglected or misinterpreted by many economic agents. Therefore, a greater willingness to delegate the decision to the robo advisor presumably leads to greater investment success or higher compensation (Hypothesis 2). The fact that is most often neglected is probably the correlation of the returns of different shares (Hypothesis 3).

Although the subjects know that the robo advisor optimally evaluates all relevant information and makes the best possible diversification decision in each case, experience has shown that many economic subjects are reluctant to entrust themselves to an algorithm— in this case a robo advisor (Hypothesis 1). Thus, if robo advisors are to be successfully established, measures to mitigate algorithm aversion have to be considered. One possible measure would be to place the decision to use a robo advisor in the context of decision for others. After all, investment decisions are not only important for the wealthy person but also for his or her family, especially children and grandchildren. Thus, in the case of decision for others, the willingness to use the robo advisor might increase (Hypothesis 4) because economic agents might try harder to make a meaningful decision when making decisions that (also) affect others (Hypothesis 5).

## 4. Results

Of the 160 participants, 80 subjects played the treatment 'Self' and 80 played the treatment 'Representative'. The experiment was carried out using z-Tree (Fischbacher 2007). The time needed for reading the instructions of the experiment (Appendix A), answering the test questions (Appendix B), and carrying out the four tasks took 15 min on average. An average payment of EUR 6.89 seemed very attractive for the amount of time required. It was intended to be sufficient incentive for meaningful economic decisions, and the subjects did actually give the impression of being concentrated and motivated.

In the first instance, it could be seen that algorithm aversion occurred to a considerable extent. Although it was clear to all of the participants that using the algorithm (robo advisor) definitely led to the best possible decisions, the robo advisor was deployed in less than half of the cases. A total of 160 subjects had to make four decisions each. This was a total of 640 decisions. The subjects decided to delegate the task to the robo advisor in only 258 cases (40.31%). In 382 cases (59.69%), the subjects refrained from using the algorithm (Figure 2). The reason why this is so remarkable is that all of the subjects knew that the robo advisor evaluated all of the relevant data in an optimal way and therefore always made the best possible decision.

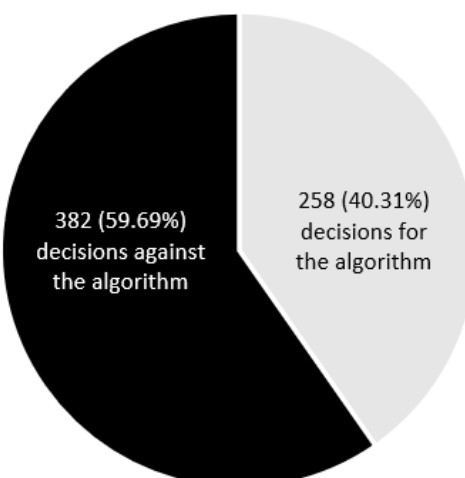

**Figure 2.** Decisions for and against the algorithm (robo advisor).

An average subject relied on the algorithm in only 1.612 out of 4 rounds. The *t*-test shows in all clarity that Null Hypothesis 1 has to be rejected (*p*-value = 0.000). The Z-test supports that only very few subjects (36 out of 160) consistently followed the rational strategy and relied on the algorithm in all rounds of the experiment (*p*-value = 0.000). Algorithm aversion thus obviously occurred to a considerable extent (59.69% of all decisions).

It is of particular interest whether this tendency towards algorithm aversion really led to a smaller number of optimal diversification decisions and whether the payments were lower than would have been the case when the subjects had consistently trusted the robo advisor. After all, one cannot simply presume that the decisions of the subjects who did not always use the robo advisor were really less successful.

A total of 53 subjects did not delegate their decision to the robo advisor a single time. In 89 out of 212 decisions (41.98%), these subjects selected optimal portfolios. On average, they achieved an expected payment value of EUR 6.36. How much the actual payment was also depended on the specific random events (dividend payments). Here, there was an average payment of EUR 6.67 (Table 7).

**Table 7.** Average success in relation to the extent of algorithm aversion.

| Number of Times the Algorithm Was Chosen | Number of Subjects | Optimal Portfolios | Expected Value of the Payment in Euros | Actual Payment in Euros |
|---|---|---|---|---|
| 0 | 53 | 89 (41.98%) | EUR 6.36 | EUR 6.67 |
| 1 | 39 | 71 (45.51%) | EUR 6.20 | EUR 6.49 |
| 2 | 19 | 51 (67.11%) | EUR 7.23 | EUR 6.78 |
| 3 | 15 | 48 (80.00%) | EUR 7.37 | EUR 7.46 |
| 4 | 34 | 136 (100%) | EUR 8.13 | EUR 7.53 |

A total of 34 subjects delegated all four of their decisions to the robo advisor. As was to be expected, in 136 out of 136 decisions (100%), the optimal portfolios were chosen. The subjects achieved an expected payment value of EUR 8.13. The specific random events (dividend payments) led to an average payment of EUR 7.53 (Table 7).

Figure 3 shows clearly that the more frequently the subjects delegated their decision to the robo advisor, the more successful they were. The subjects who did not put their faith in the robo advisor a single time achieved an average of only 1.68 optimal portfolios. The subjects who used the robo advisor to solve all four tasks made 4.00 optimal decisions. The F-test confirms: the more frequently the robo advisor was used, the more optimal portfolios were compiled (thick grey line, left scale, *p*-value = 0.000), the higher the expected value of

payment (dashed black line, right scale, *p*-value = 0.000), and the higher the actual payment (continuous black line, right scale, *p*-value = 0.000).

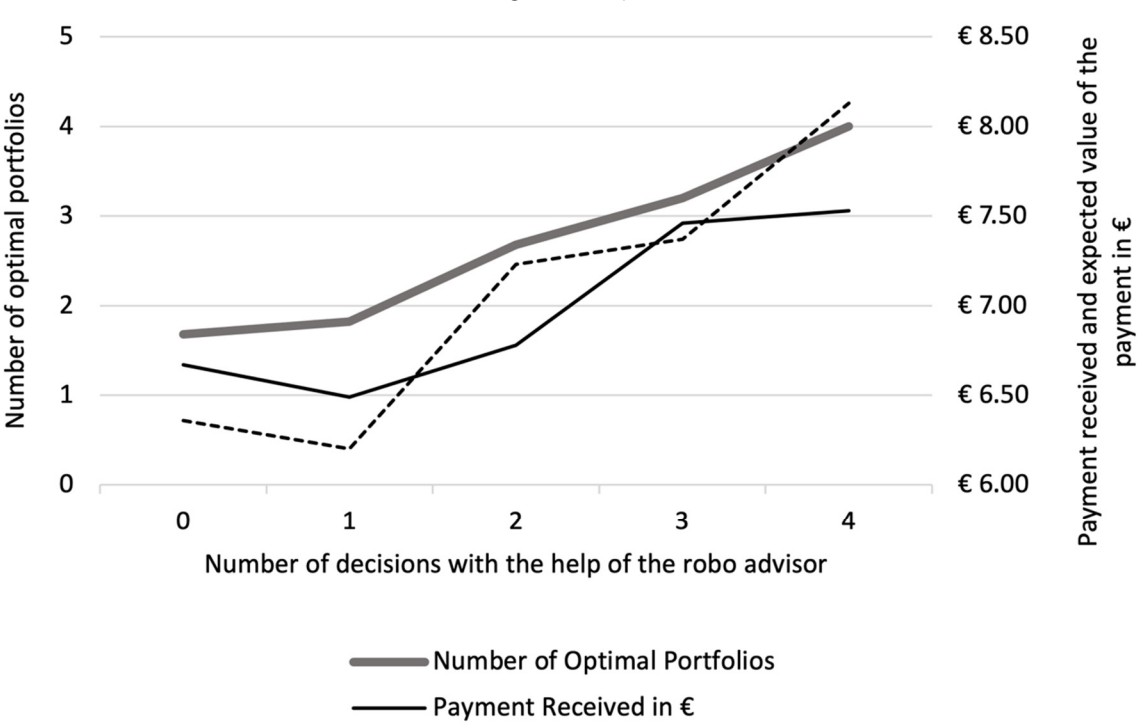

**Figure 3.** Average success in relation to the extent of algorithm aversion.

The stronger the effect of algorithm aversion, the less successful the subjects were. Null Hypothesis 2 thus has to be discarded.

Now let us look at the success of the decisions which were not delegated to the robo advisor. Tasks 1 and 2 can be solved well with the simple understanding of the diversification of the 1/n heuristic. In tasks 3 and 4, however, it is absolutely necessary to take the correlations between the dividend payments of the two shares into account and to understand the variances of the dividend payments of the two shares. Among the decisions which are not delegated to the robo advisor, a clear difference can indeed be seen between the success rate in tasks 1 and 2 on the one hand and the success rates in tasks 3 and 4 on the other. In tasks 1 and 2, 81 out of 182 decisions (44.51%) led to optimal portfolios. In tasks 3 and 4, on the other hand, only 56 out of 200 decisions (28%) led to optimal portfolios which maximised the expected value of the payment. In the chi square test, this difference proves to be significant (*p*-value = 0.001). Null Hypothesis 3 thus has to be rejected (Figure 4).

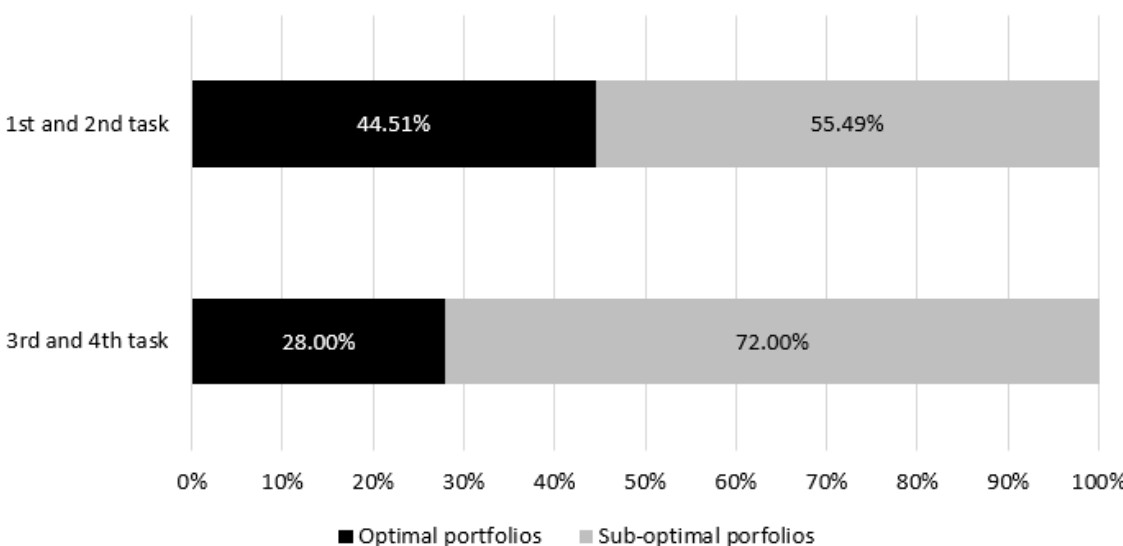

**Figure 4.** Percentage share of optimal portfolios according to tasks.

In a comparison of the two treatments 'Self' and 'Representative', no noteworthy differences with regard to use of the robo advisor can be seen. In the treatment 'Self', 131 of out 320 decisions (40.94%) were delegated to the robo advisor. In the treatment 'Representative', 127 out of 320 decisions (39.69%) were delegated to the robo advisor (Table 8 and Figure 5). This is only a very small difference. It proves to be insignificant both in the Wilcoxon rank sum test (*p*-value = 0.752) as well as in the chi square test (*p*-value = 0.747). Null Hypothesis 4 can therefore not be rejected.

**Table 8.** Influence of the treatments on algorithm aversion.

| Treatment | Robo Advisor | Own Decision | Total |
|---|---|---|---|
| 'Self' | 131 | 189 | 320 |
| 'Representative' | 127 | 193 | 320 |

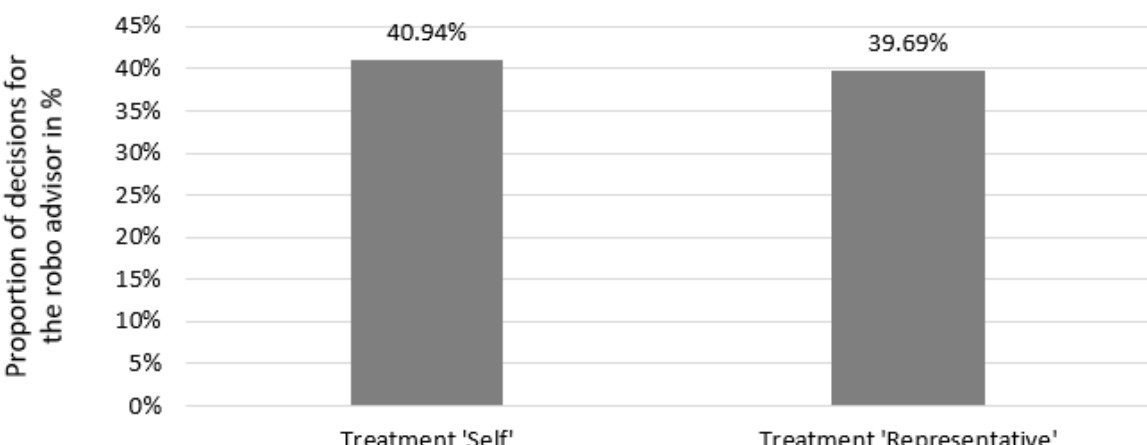

**Figure 5.** Acceptance of the robo advisor according to treatments.

This is a surprising result. The subjects in the treatment 'Representative' could have easily transferred their responsibility for the payment of another person to the robo advisor. Given that the robo advisor was known for the fact that it always made optimal decisions, nobody needed to be afraid of being criticised. However, a large part of the subjects obviously had such far-reaching reservations regarding the deployment of a robo advisor

that they did not want to take this route. We thus have to come to the conclusion that algorithm aversion occurs frequently and is by no means easy to overcome.

However, it is noticeable that it does make a difference whether one makes decisions for oneself or for others. The subjects in the treatment 'Representative' really did make a greater effort to make meaningful decisions. This can be seen in the decisions they made without using the robo advisor. In 57 out of 189 decisions (30.16%) the subjects in the treatment 'Self' succeeded in building optimal portfolios (portfolios with the highest expectation value for the payment in euros). In 80 out of 193 decisions (41.45%), the subjects in the treatment 'Representative' succeeded in building optimal portfolios (portfolios with the highest expectation value for the payment in euros) (Table 9 and Figure 6). This difference turns out to be statistically significant in the chi square test (*p*-value = 0.021).

**Table 9.** Success of portfolio decisions without the robo advisor according to treatments.

| Treatment | Number of Subjects | Number of Optimal Portfolios without the Robo Advisor | Number of Suboptimal Portfolios without the Robo Advisor | Number of Decisions Made by the Robo Advisor | Total |
|---|---|---|---|---|---|
| 'Self' | 80 | 57 | 132 | 131 | 320 |
| 'Representative' | 80 | 80 | 113 | 127 | 320 |

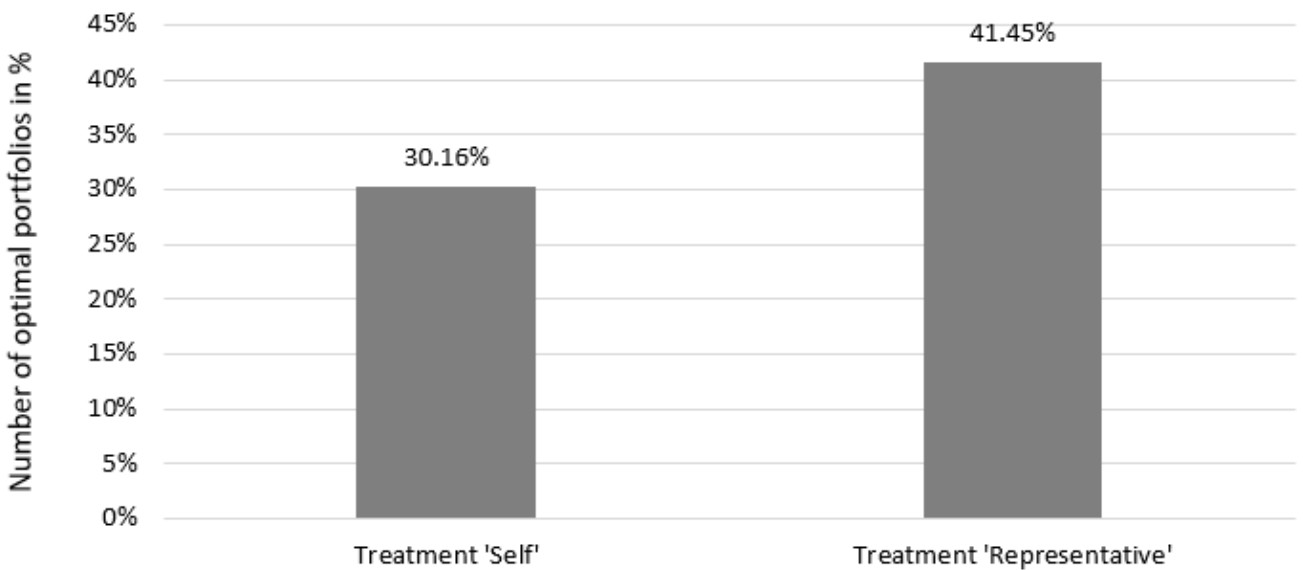

**Figure 6.** Success of the portfolio decisions without the robo advisor according to treatments.

A clear difference between the two treatments can definitely be seen. The subjects behaved differently depending on whether they were deciding for themselves or for others. They obviously acted less impulsively in the treatment 'Representative' and weighed up more precisely which portfolio composition would presumably lead to the largest payment. However, this effort to make meaningful decisions did not lead to a greater acceptance of robo advisors. The subjects' reservations about using an algorithm were obviously stronger than their wishes to make decisions for others with particular care.

## 5. Discussion

Our results contribute to the academic debate in three ways. First, it has been shown that many subjects have massive reservations about robo advisors despite their obvious advantages. In our study, robo advisors consistently outperformed subjects. Still, most

subjects chose not to use them. Although robo advisors have enormous potential and perform significantly better on average, they seemed to be very unpopular among subjects. This is in line with previous studies, which also found that algorithm aversion in particular can be a hurdle in establishing robo advisors (Hodge et al. 2021; Alemanni et al. 2020; Niszczota and Kaszás 2020).

Second, our research confirms that algorithm aversion is a serious barrier to the diffusion of innovative business fields in general. In this respect, we may also be facing a societal problem. Already today, the use of algorithms clearly provides humans with more powerful options for solving problems. Yet, decision-makers refuse to use them. Instead, they perform tasks themselves, leading to higher costs and poorer results. It therefore remains an important task of research, especially with regard to cognitive biases and heuristics, to further explore the background of algorithm aversion in order to contribute to the progress of society.

Third, it turns out that it makes little difference to the extent of algorithm aversion who has to bear the consequences (oneself or third parties). Research by Back et al. (2021) suggests that one reason to consult a robo advisor might be that it feels like relinquishing some of the responsibility for unpleasant tasks and potential mistakes. However, this assumption was not confirmed in our study. If subjects made decisions for others who may have demanded a justification for possible mistakes, the robo advisor was nevertheless just as unpopular.

To save taxes, many wealthy private clients transfer part of their assets to their children while they are still minors. These assets also need to be managed. The parents now have to decide on behalf of their children how this should be accomplished. If algorithm aversion were less prominent in decisions for others, this could be a starting point to resolve or at least mitigate the bias against robo advisors. However, no evidence for this has emerged. Algorithm aversion is reflected to the same extent in the decisions that economic agents make for themselves and in the decisions that they make for others.

Of course, there are also some limitations that may affect the validity of our results for practical applications. First, it should be mentioned that the results were obtained in the context of financial decisions with robo advisors. Financial decisions are influenced by a variety of factors, such as financial literacy or experience. Algorithm aversion is far from being the only influencing factor. It may therefore be worthwhile to revisit our research question in relation to other areas of use for algorithms.

Moreover, robo advisors from reputable banks go through a detailed accreditation process. In this process, independent experts verify, for example, whether the robo advisors take appropriate measures to hedge risks and also make decisions that are justifiable from an ethical point of view. Accreditation is thus a tool that can increase user confidence. However, it cannot be replicated in the same way in an economic laboratory experiment.

Finally, when making decisions on behalf of others, it may always make a difference what one's relationship is to the person who has to bear the consequences. We conducted a laboratory experiment at our research institution. Usually, students go there together with fellow students whom they know from classes. Sometimes students also come alone. As such, the consequences of the decision in the treatment 'Representative' were largely borne either by complete strangers or loose acquaintances. It must be left to future research efforts to see if a different outcome emerges when we decide, for example, on behalf of loved ones.

## 6. Conclusions

Robo advisors are algorithms which can automatically make investment decisions for asset management customers. Given the increased price sensitivity of wealthy private clients, robo advisors are one way to offer solid portfolio management decisions at a low cost. However, customers have considerable reservations about algorithms, even when they are very efficient systems. This phenomenon, which is known as algorithm aversion, is considered in more detail in this study.

In a laboratory experiment, subjects made a total of four portfolio decisions. They could either try to determine the optimal portfolio composition in each case themselves or they could delegate the task to a robo advisor. The robo advisor took all the relevant information into account in an optimal way and always chose the portfolio composition which led to the highest expected value of the payment in euros. The subjects were familiar with the qualities of the robo advisor. Nevertheless, they only used it in around 40% of all cases. In around 60% of all decision making situations, the subjects trusted in their own judgement, although it must have been clear to them that they were not able to make better decisions than the robo advisor. Algorithm aversion thus occurred to a great extent.

The actual success rate of the subjects who did not put their faith in the robo advisor was indeed lower than that of the robo advisor. This applied to the average number of optimal portfolio compositions, to the average expected values of their payment in euros, and also with regard to the actually obtained payment in euros. It is crystal clear that the more frequently the subjects delegated their decisions to the robo advisor, the greater their success. With their aversion towards the algorithm, the subjects were recognisably damaging themselves.

The subjects had particular difficulties when trying to take into account the correlation between the different investments. Tasks which could be solved with the simple diversification strategy of the $1/n$ heuristic (tasks 1 and 2) were dealt with successfully significantly more often than tasks which could not be suitably dealt with using the $1/n$ heuristic (tasks 3 and 4).

Ultimately, it became clear that subjects who had to make decisions for others approached the task in a more careful and concentrated way. Among the decisions which were not made by the robo advisor, there were significantly more optimal portfolios within the subjects who made decisions for others than among those who decided for themselves. However, this did not have an effect on algorithm aversion. Regardless of whether the subjects decided for themselves or for others, a readiness to delegate the decision to the robo advisor could only be seen in around 40% of decisions.

To summarise, the following can be stated: The deployment of robo advisors can, under certain circumstances, be a low-cost and very efficient alternative to traditional asset management. However, algorithm aversion hinders the establishment of the business which could be had with robo advisors.

**Supplementary Materials:** The following supporting information can be downloaded at: https://www.mdpi.com/article/10.3390/jrfm15080353/s1.

**Author Contributions:** Conceptualization, M.S.; Software, I.F., J.R.J., and M.L.; Validation, I.F., J.R.J., M.L., and M.S.; Formal analysis, I.F., J.R.J., M.L., and M.S.; Data curation, I.F., J.R.J., M.L., and M.S.; Writing—original draft preparation, J.R.J., M.L., and M.S.; Writing—review and editing, I.F., J.R.J., M.L., and M.S.; Visualization, J.R.J., and M.L. All authors have read and agreed to the published version of the manuscript.

**Funding:** This research received no external funding.

**Data Availability Statement:** Our data can be accessed at Supplementary Materials.

**Acknowledgments:** The authors thank the editor and the anonymous reviewers for their constructive comments and useful suggestions, which were very helpful to enhance the manuscript.

**Conflicts of Interest:** The authors declare no conflict of interest.

## Appendix A. Instructions for the Experiment

*Appendix A.1. Instructions (Treatment 'Self')*

You have the task of creating portfolios of shares. A portfolio of shares is a compilation of several shares.



The development of the share prices is of no concern to you, because you profit only once from the dividend payments of the shares in 2022. The dividend is the distribution of profits of a stock exchange-listed company to its shareholders.

You will receive information about how the dividend payments might turn out, and about the probabilities of different amounts of dividend. In addition, you will be shown how the dividends of the shares have developed over the last ten years.

You are paid the risk-adjusted dividend. A risk-adjusted dividend is the dividend payment divided by the variance of the dividend payments of the selected portfolio. Your task thus consists of achieving the highest possible dividends with the lowest possible risk (low variance).

The total of all risk-adjusted dividends (in ECU) which you achieve via your portfolio decisions is multiplied by five at the end and then paid in euros.

You can make the portfolio decisions yourself or delegate them to an algorithm (robo advisor). The robo advisor is specialised in making meaningful portfolio decisions and takes all of the relevant information into account in an optimal way in order to achieve risk-adjusted dividend payments which are as high as possible.

*Appendix A.2. Instructions (Treatment 'Representative')*

You have the task of creating portfolios of shares. A portfolio of shares is a compilation of several shares.

The development of the share prices is of no concern to you, because you profit only once from the dividend payments of the shares in 2022. The dividend is the distribution of profits of a stock exchange-listed company to its shareholders.

You will receive information about how the dividend payments might turn out, and about the probabilities of different amounts of dividend. In addition, you will be shown how the dividends of the shares have developed over the last ten years.

You are paid the risk-adjusted dividend. A risk-adjusted dividend is the dividend payment divided by the variance of the dividend payments of the selected portfolio. Your task thus consists of achieving the highest possible dividends with the lowest possible risk (low variance).

The total of all risk-adjusted dividends (in ECU) which you achieve via your portfolio decisions is multiplied by five at the end and then paid in euros. However, this amount is not paid to you, but to another participant. If you make successful decisions, one of the other participants will have something to be pleased about. If you make unsuccessful decisions, one of the other participants will be annoyed.

At the same time, another participant is making the decisions which determine your payment. Who has made portfolio decisions for whom will be announced at the end of the session.

So please remember why you made which decisions. The other participant might want you to justify your decisions if the results are disappointing.

You can make the portfolio decisions yourself or delegate them to an algorithm (robo advisor). The robo advisor is specialised in making meaningful portfolio decisions and takes all of the relevant information into account in an optimal way in order to achieve risk-adjusted dividend payments which are as high as possible.

**Appendix B. Test Questions**

*Appendix B.1. Test Questions (Treatment 'Self')*

What is a share portfolio?

(a)  A compilation of shares, bonds and derivative instruments.
(b)  A compilation of shares. *(correct)*
(c)  A compilation of various securities without shares.

What is a dividend?

(a)  It is the opposite of a multiplication.



(b) It is a major military unit.

(c) It is the distribution of profits by a stock exchange-listed company to its shareholders. *(correct)*

What do you profit from?

(a) From increases in the price of the shares that I choose.

(b) From the risk-adjusted dividends of the shares that I choose. *(correct)*

(c) From increases in the price of the shares that I choose, and from the dividends.

How can the algorithm (robo advisor) be deployed?

(a) I have to use the robo advisor.

(b) The robo advisor is not available to me.

(c) I have a free choice between either making the portfolio decisions myself or delegating the task to a robo advisor which is specialised in this field. *(correct)*

*Appendix B.2. Test Questions (Treatment 'Representative')*

What is a share portfolio?

(a) A compilation of shares, bonds and derivative instruments.

(b) A compilation of shares. *(correct)*

(c) A compilation of various securities without shares.

From whose decisions do you profit?

(a) From my own decisions.

(b) From the decisions of all participants.

(c) From the decisions of the participant who makes the decisions for me. *(correct)*

What determines the payment of the person for whom you make the decisions?

(a) The changes in the prices of the shares that I choose.

(b) The risk-adjusted dividends of the shares that I choose. *(correct)*

(c) The increases in the price of the shares that I choose, and the dividends of the shares that I choose.

How can the algorithm (robo advisor) be deployed?

(a) I have to use the robo advisor.

(b) The robo advisor is not available to me.

(c) I have a free choice between either making the portfolio decisions myself or delegating the task to a robo advisor which is specialised in this field. *(correct)*

**Appendix C. The Tasks**

*Appendix C.1. Task 1 (Treatment 'Self')*

There are two shares to choose from: share Y and share Z. The dividend payments of the two companies are independent random processes with two possible configurations: ECU 8 and ECU 0, and with an expected value of ECU 4. In the table you can see how high the dividend payments of the two shares were in the last 10 years.

**Table A1.** Dividend payments of the shares in task 1 of treatment 'Self'.

|  | 2012 | 2013 | 2014 | 2015 | 2016 | 2017 | 2018 | 2019 | 2020 | 2021 | 2022 |
|---|---|---|---|---|---|---|---|---|---|---|---|
| Share Y | ECU 8 | ECU 0 | ECU 8 | ECU 8 | ECU 8 | ECU 0 | ECU 8 | ECU 0 | ECU 0 | ECU 0 | ? |
| Share Z | ECU 0 | ECU 0 | ECU 8 | ECU 0 | ECU 8 | ECU 8 | ECU 0 | ECU 0 | ECU 8 | ECU 8 | ? |

You may choose two shares. As payment you receive the risk-adjusted dividends of the two selected shares. The risk-adjusted dividend corresponds to the dividend payment divided by the variance of the dividend payments of the selected portfolio. Depending on the portfolio selected, you thus receive the risk-adjusted dividends of 2 Y shares, of 2 Z shares, or of 1 Y share + 1 Z share. As the dividend payments are determined by a random

process, it is not only the content of the portfolio which determines your payment, but also luck. Which event (ECU 8 or ECU 0) occurs in the case of the two shares is determined separately by drawing lots for each round of the experimental survey.

You can make the portfolio decisions yourself or delegate them to an algorithm (robo advisor). The robo advisor is specialised in making meaningful portfolio decisions and takes all of the relevant information into account in an optimal way. However, the robo advisor also does not know which random event (ECU 8 or ECU 0) will occur as the dividend of the shares. In other words, even when the robo advisor is used, luck determines the payment to a certain extent.

Now make your choice!

○     I will let the robo advisor decide;

I will decide myself and choose:

○     2 Y shares;
○     2 Z shares;
○     1 Y share + 1 Z share.

*Appendix C.2. Task 2 (Treatment 'Self')*

There are two shares to choose from: share X and share Q. The dividend payments of the two companies are independent random processes with two possible configurations: ECU 4 and ECU 0, and with an expected value of ECU 2. In the table you can see how high the dividend payments of the two shares were in the last 10 years.

**Table A2.** Dividend payments of the shares in task 2 of treatment 'Self'.

|         | 2012  | 2013  | 2014  | 2015  | 2016  | 2017  | 2018  | 2019  | 2020  | 2021  | 2022 |
|---------|-------|-------|-------|-------|-------|-------|-------|-------|-------|-------|------|
| Share X | ECU 0 | ECU 0 | ECU 4 | ECU 0 | ECU 0 | ECU 0 | ECU 4 | ECU 4 | ECU 4 | ECU 4 | ?    |
| Share Q | ECU 0 | ECU 4 | ECU 4 | ECU 4 | ECU 0 | ECU 4 | ECU 0 | ECU 0 | ECU 4 | ECU 0 | ?    |

You may choose four shares. As payment you receive the risk-adjusted dividends of the four selected shares. The risk-adjusted dividend corresponds to the dividend payment divided by the variance of the dividend payments of the selected portfolio. Depending on the portfolio selected, you thus receive the risk-adjusted dividends of 4 X shares, of 4 Q shares, of 3 X shares + 1 Q share, of 3 Q shares + 1 X share, or of 2 X shares + 2 Q shares. As the dividend payments are determined by a random process, it is not only the content of the portfolio which determines your payment, but also luck. Which event (ECU 4 or ECU 0) occurs in the case of the two shares is determined separately by drawing lots for each round of the experimental survey.

You can make the portfolio decisions yourself or delegate them to an algorithm (robo advisor). The robo advisor is specialised in making meaningful portfolio decisions and takes all of the relevant information into account in an optimal way. However, the robo advisor also does not know which random event (ECU 4 or ECU 0) will occur as the dividend of the shares. In other words, even when the robo advisor is used, luck determines the payment to a certain extent.

Now make your choice!

○     I will let the robo advisor decide;

I will decide myself and choose:

○     4 X shares;
○     4 Q shares;
○     3 X shares + 1 Q share;
○     3 Q shares + 1 X share;
○     2 Q shares + 2 X shares.

*Appendix C.3. Task 3 (Treatment 'Self')*

There are two shares from a specific sector of industry to choose from (share K and share L). In the table you can see how high the dividend payments of the two shares were in the last 10 years. When business is good in the sector, the dividend of share K is ECU 6, and that of share L is ECU 7. When business is poor in the sector, the dividend of share K is ECU 2, and that of share L is ECU 1. The business situation in the sector can vary from year to year and thus has to be viewed as a random process: the probability of the business situation being either good or poor in 2022 is 50% in each case.

**Table A3.** Dividend payments of the shares in task 3 of treatment 'Self'.

|         | 2012  | 2013  | 2014  | 2015  | 2016  | 2017  | 2018  | 2019  | 2020  | 2021  | 2022 |
|---------|-------|-------|-------|-------|-------|-------|-------|-------|-------|-------|------|
| Share K | ECU 2 | ECU 6 | ECU 2 | ECU 6 | ECU 6 | ECU 6 | ECU 2 | ECU 6 | ECU 2 | ECU 2 | ?    |
| Share L | ECU 1 | ECU 7 | ECU 1 | ECU 7 | ECU 7 | ECU 7 | ECU 1 | ECU 7 | ECU 1 | ECU 1 | ?    |

You may choose two shares. As payment you receive the risk-adjusted dividends of the two selected shares. The risk-adjusted dividend corresponds to the dividend payment divided by the variance of the dividend payments of the selected portfolio. Depending on the portfolio selected, you thus receive the risk-adjusted dividends of 2 K shares, of 2 L shares, or of 1 K share + 1 L share. As the dividend payments are determined by a random process, it is not only the content of the portfolio which determines your payment, but also luck. Which event (good or poor economic situation in the sector) occurs in the case of the two shares is determined separately by drawing lots for each round of the experimental survey.

You can make the portfolio decisions yourself or delegate them to an algorithm (robo advisor). The robo advisor is specialised in making meaningful portfolio decisions and takes all of the relevant information into account in an optimal way. However, the robo advisor also does not know which random event (good or poor economic situation in the sector) will occur as the dividend of the shares. In other words, even when the robo advisor is used, luck determines the payment to a certain extent.

Now make your choice!

○     I will let the robo advisor decide;

I will decide myself and choose:

○     2 K shares;
○     2 L shares;
○     1 K share + 1 L share.

*Appendix C.4. Task 4 (Treatment 'Self')*

There are two shares from a specific sector of industry to choose from (share M and share P). In the table you can see how high the dividend payments of the two shares were in the last 10 years. When business is good in the sector, the dividend of share M is ECU 4, and that of share P is ECU 3. When business is poor in the sector, the dividend of share M is ECU 0, and that of share P is ECU 1. The business situation in the sector can vary from year to year and thus has to be viewed as a random process: the probability of the business situation being either good or poor in 2022 is 50% in each case.

**Table A4.** Dividend payments of the shares in task 4 of treatment 'Self'.

|         | 2012  | 2013  | 2014  | 2015  | 2016  | 2017  | 2018  | 2019  | 2020  | 2021  | 2022 |
|---------|-------|-------|-------|-------|-------|-------|-------|-------|-------|-------|------|
| Share M | ECU 4 | ECU 0 | ECU 4 | ECU 0 | ECU 0 | ECU 0 | ECU 4 | ECU 4 | ECU 0 | ECU 4 | ?    |
| Share P | ECU 3 | ECU 1 | ECU 3 | ECU 1 | ECU 1 | ECU 1 | ECU 3 | ECU 3 | ECU 1 | ECU 3 | ?    |

You may choose four shares. As payment you receive the risk-adjusted dividends of the four selected shares. The risk-adjusted dividend corresponds to the dividend payment divided by the variance of the dividend payments of the selected portfolio. Depending on the portfolio selected, you thus receive the risk-adjusted dividends of 4 M shares, of 4 P shares, of 3 M shares + 1 P share, of 3 P shares + 1 M share, or of 2 M shares + 2 P shares. As the dividend payments are determined by a random process, it is not only the content of the portfolio which determines your payment, but also luck. Which event (good or poor economic situation in the sector) occurs in the case of the two shares is determined separately by drawing lots for each round of the experimental survey.

You can make the portfolio decisions yourself or delegate them to an algorithm (robo advisor). The robo advisor is specialised in making meaningful portfolio decisions and takes all of the relevant information into account in an optimal way. However, the robo advisor also does not know which random event (good or poor economic situation in the sector) will occur as the dividend of the shares. In other words, even when the robo advisor is used, luck determines the payment to a certain extent.

Now make your choice!

○     I will let the robo advisor decide;

I will decide myself and choose:

○     4 M shares;
○     4 P shares;
○     3 M shares + 1 P share;
○     3 P shares + 1 M share;
○     2 M shares + 2 P shares.

*Appendix C.5. Task 1 (Treatment 'Representative')*

There are two shares to choose from: share Y and share Z. The dividend payments of the two companies are independent random processes with two possible configurations: ECU 8 and ECU 0, and with an expected value of ECU 4. In the table you can see how high the dividend payments of the two shares were in the last 10 years.

**Table A5.** Dividend payments of the shares in task 1 of treatment 'Representative'.

|  | 2012 | 2013 | 2014 | 2015 | 2016 | 2017 | 2018 | 2019 | 2020 | 2021 | 2022 |
|---|---|---|---|---|---|---|---|---|---|---|---|
| Share Y | ECU 8 | ECU 0 | ECU 8 | ECU 8 | ECU 8 | ECU 0 | ECU 8 | ECU 0 | ECU 0 | ECU 0 | ? |
| Share Z | ECU 0 | ECU 0 | ECU 8 | ECU 0 | ECU 8 | ECU 8 | ECU 0 | ECU 0 | ECU 8 | ECU 8 | ? |

You may choose two shares. As compensation, the risk-adjusted dividends are paid from the two selected shares. The risk-adjusted dividend corresponds to the dividend payment divided by the variance of the dividend payments of the selected portfolio. Depending on the portfolio selection, the risk-adjusted dividend of 2 Y shares, of 2 Z shares, or of 1 Y share + 1 Z share is paid out. As the dividend payments are determined by a random process, it is not only the content of the portfolio which determines your payment, but also luck. Which event (ECU 8 or ECU 0) occurs in the case of the two shares is determined separately by drawing lots for each round of the experimental survey.

You can make the portfolio decisions yourself or delegate them to an algorithm (robo advisor). The robo advisor is specialised in making meaningful portfolio decisions and takes all of the relevant information into account in an optimal way. However, the robo advisor also does not know which random event (ECU 8 or ECU 0) will occur as the dividend of the shares. In other words, even when the robo advisor is used, luck determines the payment to a certain extent.

The payment which you achieve with your decision is received by one of the other participants and not by you. This other participant might ask you to justify your choices, so you should think carefully about the decisions you make.

Now make your choice!

○    I will let the robo advisor decide;

I will decide myself and choose:

○    2 Y shares;
○    2 Z shares;
○    1 Y share + 1 Z share.

*Appendix C.6. Task 2 (Treatment 'Representative')*

There are two shares to choose from: share X and share Q. The dividend payments of the two companies are independent random processes with two possible configurations: ECU 4 and ECU 0, and with an expected value of ECU 2. In the table you can see how high the dividend payments of the two shares were in the last 10 years.

**Table A6.** Dividend payments of the shares in task 2 of treatment 'Representative'.

| | 2012 | 2013 | 2014 | 2015 | 2016 | 2017 | 2018 | 2019 | 2020 | 2021 | 2022 |
|---|---|---|---|---|---|---|---|---|---|---|---|
| Share X | ECU 0 | ECU 0 | ECU 4 | ECU 0 | ECU 0 | ECU 0 | ECU 4 | ECU 4 | ECU 4 | ECU 4 | ? |
| Share Q | ECU 0 | ECU 4 | ECU 4 | ECU 4 | ECU 0 | ECU 4 | ECU 0 | ECU 0 | ECU 4 | ECU 0 | ? |

You may choose four shares. As compensation, the risk-adjusted dividends are paid from the four selected shares. The risk-adjusted dividend corresponds to the dividend payment divided by the variance of the dividend payments of the selected portfolio. Depending on the portfolio selection, the risk-adjusted dividend of 4 X shares, of 4 Q shares, of 3 X shares + 1 Q share, of 3 Q shares + 1 X share, or of 2 X shares + 2 Q shares is paid out. As the dividend payments are determined by a random process, it is not only the content of the portfolio which determines your payment, but also luck. Which event (ECU 4 or ECU 0) occurs in the case of the two shares is determined separately by drawing lots for each round of the experimental survey.

You can make the portfolio decisions yourself or delegate them to an algorithm (robo advisor). The robo advisor is specialised in making meaningful portfolio decisions and takes all of the relevant information into account in an optimal way. However, the robo advisor also does not know which random event (ECU 4 or ECU 0) will occur as the dividend of the shares. In other words, even when the robo advisor is used, luck determines the payment to a certain extent.

The payment which you achieve with your decision is received by one of the other participants and not by you. This other participant might ask you to justify your choices, so you should think carefully about the decisions you make.

Now make your choice!

○    I will let the robo advisor decide;

I will decide myself and choose:

○    4 X shares;
○    4 Q shares;
○    3 X shares + 1 Q share;
○    3 Q shares + 1 X share;
○    2 Q shares + 2 X shares.

*Appendix C.7. Task 3 (Treatment 'Representative')*

There are two shares from a specific sector of industry to choose from (share K and share L). In the table you can see how high the dividend payments of the two shares were in the last 10 years. When business is good in the sector, the dividend of share K is ECU 6, and that of share L is ECU 7. When business is poor in the sector, the dividend of share K is ECU 2, and that of share L is ECU 1. The business situation in the sector can vary from

year to year and thus has to be viewed as a random process: the probability of the business situation being either good or poor in 2022 is 50% in each case.

**Table A7.** Dividend payments of the shares in task 3 of treatment 'Representative'.

|  | 2012 | 2013 | 2014 | 2015 | 2016 | 2017 | 2018 | 2019 | 2020 | 2021 | 2022 |
|---|---|---|---|---|---|---|---|---|---|---|---|
| Share K | ECU 2 | ECU 6 | ECU 2 | ECU 6 | ECU 6 | ECU 6 | ECU 2 | ECU 6 | ECU 2 | ECU 2 | ? |
| Share L | ECU 1 | ECU 7 | ECU 1 | ECU 7 | ECU 7 | ECU 7 | ECU 1 | ECU 7 | ECU 1 | ECU 1 | ? |

You may choose two shares. As compensation, the risk-adjusted dividends are paid from the two selected shares. The risk-adjusted dividend corresponds to the dividend payment divided by the variance of the dividend payments of the selected portfolio. Depending on the portfolio selection, the risk-adjusted dividend of 2 K shares, of 2 L shares, or of 1 K share + 1 L share is paid out. As the dividend payments are determined by a random process, it is not only the content of the portfolio which determines your payment, but also luck. Which event (good or poor economic situation in the sector) occurs in the case of the two shares is determined separately by drawing lots for each round of the experimental survey.

You can make the portfolio decisions yourself or delegate them to an algorithm (robo advisor). The robo advisor is specialised in making meaningful portfolio decisions and takes all of the relevant information into account in an optimal way. However, the robo advisor also does not know which random event (good or poor economic situation in the sector) will occur as the dividend of the shares. In other words, even when the robo advisor is used, luck determines the payment to a certain extent.

The payment which you achieve with your decision is received by one of the other participants and not by you. This other participant might ask you to justify your choices, so you should think carefully about the decisions you make.

Now make your choice!

○    I will let the robo advisor decide;

I will decide myself and choose:

○    2 K shares;
○    2 L shares;
○    1 K share + 1 L share.

*Appendix C.8. Task 4 (Treatment 'Representative')*

There are two shares from a specific sector of industry to choose from (share M and share P). In the table you can see how high the dividend payments of the two shares were in the last 10 years. When business is good in the sector, the dividend of share M is ECU 4, and that of share P is ECU 3. When business is poor in the sector, the dividend of share M is ECU 0, and that of share P is ECU 1. The business situation in the sector can vary from year to year and thus has to be viewed as a random process: the probability of the business situation being either good or poor in 2022 is 50% in each case.

**Table A8.** Dividend payments of the shares in task 4 of treatment 'Representative'.

|  | 2012 | 2013 | 2014 | 2015 | 2016 | 2017 | 2018 | 2019 | 2020 | 2021 | 2022 |
|---|---|---|---|---|---|---|---|---|---|---|---|
| Share M | ECU 4 | ECU 0 | ECU 4 | ECU 0 | ECU 0 | ECU 0 | ECU 4 | ECU 4 | ECU 0 | ECU 4 | ? |
| Share P | ECU 3 | ECU 1 | ECU 3 | ECU 1 | ECU 1 | ECU 1 | ECU 3 | ECU 3 | ECU 1 | ECU 3 | ? |

You may choose four shares. As compensation, the risk-adjusted dividends are paid from the four selected shares. The risk-adjusted dividend corresponds to the dividend payment divided by the variance of the dividend payments of the selected portfolio.

Depending on the portfolio selection, the risk-adjusted dividend of 4 M shares, of 4 P shares, of 3 M shares + 1 P share, of 3 P shares + 1 M share, or of 2 M shares + 2 P shares is paid out. As the dividend payments are determined by a random process, it is not only the content of the portfolio which determines your payment, but also luck. Which event (good or poor economic situation in the sector) occurs in the case of the two shares is determined separately by drawing lots for each round of the experimental survey.

You can make the portfolio decisions yourself or delegate them to an algorithm (robo advisor). The robo advisor is specialised in making meaningful portfolio decisions and takes all of the relevant information into account in an optimal way. However, the robo advisor also does not know which random event (good or poor economic situation in the sector) will occur as the dividend of the shares. In other words, even when the robo advisor is used, luck determines the payment to a certain extent.

The payment which you achieve with your decision is received by one of the other participants and not by you. This other participant might ask you to justify your choices, so you should think carefully about the decisions you make.

Now make your choice!

○    I will let the robo advisor decide;

I will decide myself and choose:

○    4 M shares;
○    4 P shares;
○    3 M shares + 1 P share;
○    3 P shares + 1 M share;
○    2 M shares + 2 P shares.

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
