# Peer review of "Algorithm Aversion as an Obstacle in the Establishment of Robo Advisors"

_jrfm, doi:10.3390/jrfm15080353_

Round 1
Reviewer 1 Report
The paper is interesting and deals with a very topical issue, however there are a number of problems that need to be solved:
There is no clear research question.
The hypotheses are not related to that question and there is no reference to them in the conclusions.
However, the explanation of the experiment is somewhat confusing, it is recommended to reformulate it and use some kind of visual aid.
The experiment is explained before explaining the composition of the sample.
Some paragraphs are not related to the rest, for example paragraph 2, page 3 "In the meantime,...".
In some cases strings of quotations are included without explanation, for example page 2 or page 3 final paragraphs.
Some articles that talk specifically about roboadvisors and their perception by humans are not included, for example:
http://dx.doi.org/10.3390/joitmc5040081
Finally, the conclusions section is undeveloped, lacking theoretical and managerial implications.
Author Response
We are very grateful that the reviewers approached our study with patience, diligence, and great expertise. We appreciate their helpful comments and suggestions. We have spent the last days working very hard to improve our manuscript, always with the goal in mind to resubmit in time for the special issue on “Advanced Portfolio Optimization and Management”. The proposed changes have been implemented as follows:

Reviewer 2 Report
Thank you very much for having a chance to review the paper. Please find my comments below:
1. The literature review section should be improved. Additionally, the Introduction lack the proper clarification of the study's contribution and the structure of the paper.
2. The authors should take into account the issue of subjects’ attitude towards risk. Do the authors believe that the Robo advisor should always choose the option with the highest expected value (EV), or should the choice depend on the client's attitude towards risk? The study should have a direct relation to the microeconomic theory regarding risk aversion.
3. The paper lacks a discussion section. Are the results in line with the other studies? The results need to be presented fully and interpreted with the direct link of the existing literature and theories.
4. The conclusion section should be corrected. The authors should the significance and relevance of the results, limitations of the study and the plan for future research on the topic.
Author Response
We are very grateful that the reviewers approached our study with patience, diligence, and great expertise. We appreciate their helpful comments and suggestions. We have spent the last days working very hard to improve our manuscript, always with the goal in mind to resubmit in time for the special issue on “Advanced Portfolio Optimization and Management”. The proposed changes have been implemented as follows:
Reviewer 2 (1): The literature review section should be improved. Additionally, the Introduction lack the proper clarification of the study's contribution and the structure of the paper.
Response: We have made adjustments:
Literature (page 3):
We build on studies that have examined what influences willingness to use a robo advisor. Alemanni et al. (2020) show that the willingness to follow a robo advisor is lower when the robo advisor suggests a portfolio change. If the current portfolio is to be retained, the willingness to use is similar to advice from human advisors (Alemanni et al., 2020). In a questionnaire-based study, von Walter, Kremmel & Jäger (2021) find that consumers who believe artificial intelligence is better than human intelligence are more likely to accept advice from a robo advisor. Hodge, Mendoza & Sinha (2021) show that subjects follow advice from a robo advisor without a name more closely than advice from a robo advisor that has been given a name. Robo advisors with names tend to be more popular for simple tasks than for complex ones (Hodge, Mendoza & Sinha, 2021). The age of the decision maker may also be a factor: Robillard (2018) argues that Millennials may rely more heavily on robo advisors because this generation has lower trust in fellow humans than other generations.
Users' risk attitudes and attitudes toward automated processes also influence robo advisors and their investment decisions. Robo advisors can identify different risk profiles of their users, although there are large differences in risk preferences within the same investor type group (Boreiko & Massarotti, 2020). User preferences, however, have different effects on the perception and intention to use robo advisors. For financial investments, a higher perceived level of automation leads to higher performance expectations and higher user control leads to lower perceived risk (Rühr, Berger & Hess, 2019). Since robo advisors should take user preferences into account to increase their usage intent, a performance-control dilemma arises that needs to be mitigated (Rühr, 2020).
Another important aspect seems to be the transfer of responsibility to the robo advisor. Niszczota and Kaszas (2020) discovered that moral investment decisions are rather delegated to humans than to robo advisors. On the other hand, Back, Morana & Spann (2021) show that subjects feel better in case of losses if they have delegated some of the responsibility to the robo advisor. For tasks outside the world of finance, it has already been shown that punishment can be significantly lower if errors are committed by an artificial agent rather than by a subject (Feier, Gogoll & Uhl, 2022). The idea that, under certain circumstances, subjects may be happy to hand over responsibility for possible future mistakes to a robo advisor is remarkable, and we will explore it in more detail in the present study.
Structure of the paper (end of last paragraph Chapter 1, page 4):
The rest of this research paper is organized as follows: In Chapter 2, the experimental design is explained. Chapter 3 deals with the elaboration of the hypotheses. In Chapter 4 and 5, the results of the economic experiments are presented and discussed in the context of the existing literature. To wrap up this study, Chapter 6 provides a summary of the key findings and a conclusion.
Reviewer 2 (2): The authors should take into account the issue of subjects’ attitude towards risk. Do the authors believe that the Robo advisor should always choose the option with the highest expected value (EV), or should the choice depend on the client's attitude towards risk? The study should have a direct relation to the microeconomic theory regarding risk aversion.
Response: For the assessment of portfolio decisions, the risk preference of the relevant economic agent is generally of key importance. This is an aspect we have dealt with very thoroughly (see Filiz et al., 2020). However, the established methods for measuring risk preference (e.g., Holt and Laury, 2002; Eckel and Grossman, 2008; Crosetto and Filippin, 2013) have significant weaknesses. Careful elicitation of risk preference is a challenging, costly, and time-consuming endeavor (Filiz et al., 2020). Therefore, we decided to design the compensation structure in the present experiment in such a way that elicitation of risk preference is not necessary. Participants' compensation is based on risk-adjusted dividend payments. In all four tasks, the compensation for the minimum-variance portfolio is always highest, regardless of the characteristics of the random events. This applies to the actual compensation and not only to the expected value of the compensation. A decision against the minimum variant portfolio does not lead to an increase in compensation. Therefore, in this experiment, the minimum-variance portfolio is always the choice of a rational economic agent - regardless of the degree of risk preference.
References
Crosetto, P., & Filippin, A. (2013), The ‘bomb’ risk elicitation task, Journal of Risk and Uncertainty, 47(1), 31-65.
Eckel, C. C., & Grossman, P. J. (2008), Forecasting risk attitudes: an experimental study using actual and forecast gamble choices, Journal of Economic Behavior and Organization, 68(1), 1-17.
Filiz, I., Nahmer, T., Spiwoks, M., & Gubaydullina, Z. (2020), Measurement of Risk Preference, Journal of Behavioral and Experimental Finance, 27, 1-12.
Holt, C. A., & Laury, S. K. (2002), Risk aversion and incentive effects, American Economic Review, 92(5), 1644-1655.
We now also present that we pay the risk-adjusted dividend as compensation to our subjects for this reason (first paragraph, page 4):
The subjects receive the risk-adjusted return of their investment decisions as payment. This has the advantage that the subjects' risk preferences have no meaning for the assessment of the investment alternatives.
Reviewer 2 (3): The paper lacks a discussion section. Are the results in line with the other studies? The results need to be presented fully and interpreted with the direct link of the existing literature and theories.
Response: We have taken this comment on board and used it as an opportunity to add a discussion chapter. This is now presented as the 5th chapter (pages 17-18):
- Discussion
Our results contribute to the academic debate in three ways. First, it has been shown that many subjects have massive reservations about robo advisors despite their obvious advantages. In our study, robo advisors consistently outperform subjects. Still, most subjects choose not to use them. Although robo advisors have enormous potential and perform significantly better on average, they seem to be very unpopular among subjects. This is in line with previous studies, which also found that algorithm aversion in particular can be a hurdle in establishing robo advisors (Hodge, Mendoza & Sinha, 2021; Alemanni et al., 2020; Niszczota & Kaszas, 2020).
Second, our research confirms that algorithm aversion is a serious barrier to the diffusion of innovative business fields in general. In this respect, we may also be facing a societal problem. Already today, the use of algorithms clearly provides humans with more powerful options for solving problems. Yet decision makers refuse to use them. Instead, they perform tasks themselves, leading to higher costs and poorer results. It therefore remains an important task of research, especially with regard to cognitive biases and heuristics, to further explore the background of algorithm aversion in order to contribute to the progress of society.
Third, it turns out that it makes little difference to the extent of algorithm aversion who has to bear the consequences (oneself or third parties). Research by Back, Morana & Spann (2021) had suggested that one reason to consult a robo advisor might be that it feels like relinquishing some of the responsibility for unpleasant tasks and potential mistakes. However, this assumption was not confirmed in our study. If subjects make decisions for others who may demand a justification for possible mistakes, the robo advisor is nevertheless just as unpopular.
To save taxes, many wealthy private clients transfer part of their assets to their children while they are still minors. These assets also need to be managed. The parents now have to decide on behalf of their children how this should be done. If algorithm aversion had been less prominent in proxy decisions, this would have been a starting point to resolve or at least mitigate the bias against robo advisors. However, no evidence for this has emerged. Algorithm aversion is reflected to the same extent in decisions that economic agents make for themselves and in decisions that they make for others.
Reviewer 2 (4): The conclusion section should be corrected. The authors should the significance and relevance of the results, limitations of the study and the plan for future research on the topic.
Response: We show the relevance of the results in the new Discussion chapter (see previous quote). We also added limitations and future research in the end of the new discussion chapter (page 18):
Of course, there are also some limitations that may affect the validity of our results for practical applications. First, it should be mentioned that the results were obtained in the context of financial decisions with robo advisors. Financial decisions are influenced by a variety of factors, such as financial literacy or experience. Algorithm aversion is far from being the only influencing factor. It may therefore be worthwhile to revisit our research question in relation to other areas of use for algorithms.
Moreover, robo advisors from reputable banks go through a detailed accreditation process. In this process, independent experts verify, for example, whether the robo advisors take appropriate measures to hedge risks and also make decisions that are justifiable from an ethical point of view. Accreditation is thus a tool that ca n increase user confidence. However, it cannot be replicated in the same way in an economic laboratory experiment.
Finally, when making decisions on behalf of others, it may always make a difference what one's relationship is to the person who has to bear the consequences. We conducted a laboratory experiment at our research institution. Usually, students go there together with fellow students who they know from classes. Sometimes students also come alone. As such, the consequences of the decision in the treatment ‘Representative’ were largely borne either by complete strangers or loose acquaintances. It must be left to future research efforts to see if a different outcome emerges when we decide, for example, on behalf of loved ones.
We appreciate your time and effort and would like to thank the reviewers for their suggestions which have helped us improve our work. We are glad to see how our manuscript has developed and hope that you find it suitable for publication in the special issue on “Advanced Portfolio Optimization and Management”. Should you find our paper in need of additional correction, we are of course prepared to make further changes.
Round 2
Reviewer 1 Report
Congratulations for your job!!!